



# Contrasting watershed-scale trends in runoff and sediment
# yield complicate rangeland water resources planning
**M. D. Berg[1,*], F. Marcantonio[2], M. A. Allison[3,4], J. McAlister[5], B. P. Wilcox[1], and**
**W. E. Fox[1,5]**
[1]{Texas A&M University Department of Ecosystem Science and Management, College
Station, Texas}
[2]{Texas A&M University Department of Geology and Geophysics, College Station, Texas}
[3]{The Water Institute of the Gulf, Baton Rouge, Louisiana}
[4]{Tulane University Department of Earth and Environmental Sciences, New Orleans,
Louisiana}
[5]{Texas A&M AgriLife Blackland Research & Extension Center, Temple, Texas}
[*]{now at: Save Water Co, Houston, Texas}
Correspondence to: M. D. Berg (mbergtamu@gmail.com)
**Abstract**
Rangelands cover a large portion of the earth's land surface and are undergoing dramatic
landscape changes. At the same time, these ecosystems face increasing expectations to meet
growing water supply needs. To address major gaps in our understanding of rangeland
hydrologic function, we investigated historical watershed-scale runoff and sediment yield in a
dynamic landscape in central Texas, USA. We quantified the relationship between
precipitation and runoff and analyzed reservoir sediment cores dated using Cesium-137 and
Lead-210 radioisotopes. Local rainfall and streamflow showed no directional trend over a
period of 85 years, resulting in a rainfall-runoff ratio that has been resilient to watershed
changes. Reservoir sedimentation rates generally were higher before 1963, but have been
much lower and very stable since that time. Our findings suggest that (1) rangeland water
yields may be stable over long periods despite dramatic landscape changes while (2) these
same landscape changes influence sediment yields that impact downstream reservoir storage.



Relying on rangelands to meet water needs demands an understanding of how these dynamic
landscapes function and a quantification of the physical processes at work.

## 4    1    Introduction

Diverse rangeland ecosystems falling along a grassland–forest continuum cover roughly half
of the earth's land surface (Breshears, 2006). Generally precipitation-limited, they are
typically used for livestock grazing and harvesting of woody products rather than crop
production. But rangelands worldwide face numerous challenges, including (1) conversion to
urban development or cultivation; (2) shifting plant cover, such as encroachment by woody
plants and invasion by non-native species; and (3) demands for increased production without
sacrificing sustainability (Tilman et al., 2002;Van Auken, 2000;Wilcox et al., 2012b).
As growing populations look to these dynamic landscapes to provide critical ecosystem
services—including water supply and water storage—their ability to keep pace with these
demands is uncertain (Havstad et al., 2007;Jackson et al., 2001). Some of this uncertainty is
due to the tremendous variability of runoff and erosion through time and space, which can
vary by orders of magnitude even between portions of a single small field (Gaspar et al.,
2013;Ritchie et al., 2005). Landscape changes affect these processes further still; and water
and sediment yields depend on interactions between climate, vegetation, and local geology.
These complex interactions make predictions difficult; and the influence of human activity
adds yet another compounding layer of difficulty (Peel, 2009;Boardman, 2006;Vorosmarty
and Sahagian, 2000). As a result, major gaps remain in our understanding of rangeland
ecosystems. Further interdisciplinary study is imperative to develop a coherent picture of the
linkages between hydrological, ecological, and geological processes (Newman, 2006;Wilcox
and Thurow, 2006).
Some rangeland investigations have focused on the potential of these landscapes to provide
augmented water yields or storage in small reservoirs. Economic and modeling studies have
identified vegetation management as a possible means of increasing runoff and streamflow
(Griffin and McCarl, 1989;Afinowicz et al., 2005), and government agencies have
incorporated these goals into their programs (Texas State Soil and Water Conservation Board,
2005;USDA-NRCS, 2006). Other concerns center on sediment yield, which threatens
downstream surface water storage (Bennett et al., 2002;Dunbar et al., 2010). To determine





how to respond to these issues and whether related investments are worthwhile, we must gain
a better understanding of how rangeland systems function with respect to water resources.
To date, most research has been based on extrapolation of findings from relatively small-scale
studies to larger scales or on modeled results. However, because runoff and sediment
production are scale-dependent processes, such extrapolation is often unreliable (de Vente and
Poesen, 2005;Wilcox et al., 2003). Since they more accurately reveal the true water and
sediment yields of watersheds, studies of these processes conducted at the catchment scale are
much more relevant to water planning efforts. But whereas catchment-scale data on
precipitation and streamflow are somewhat widely available, corresponding sediment data are
lacking. Since they serve as archives of historical watershed conditions, the use of reservoir
sediments provides one means of filling this data gap and of investigating the impact of
human activity (Edwards and Whittington, 2001;Winter et al., 2001). Linking the findings of
such investigations with observed changes at the watershed scale will greatly facilitate the
development of effective strategies for managing rangeland water resources.
In this study, we investigated the hydrological and sediment transport dynamics of rangeland
watersheds. Our main objectives were to (1) quantify long-term trends in precipitation and
streamflow using historical data; (2) estimate historical sedimentation rates through
radioisotope analysis of reservoir sediment cores; and (3) explore the potential effects of
drought conditions on sediment production with historical data. Addressing these objectives
not only improves our understanding of rangeland processes but also provides much-needed
information on the potential of these landscapes to provide for growing global water needs.
**2    Methods**
**2.1    Study area**
As part of a broader study of landscape change and ecosystem function, we examined
rangeland processes in the Lampasas Cut Plain of central Texas, USA. This savanna
landscape is characterized by low buttes and mesas separated by broad, flat valleys. Local
prevailing geology is Cretaceous limestone; soils are loamy and clayey, with occasional sandy
loams, and are susceptible to sheet and gully erosion (Allison, 1991;Clower, 1980). The area
is drained by the Lampasas River. Streamflow in the upper reaches of the river is runoff-
dominated, with localized contributions from springflow (Prcin et al., 2013), and has been



recorded at two primary stations (Figure 1). Annual precipitation averages approximately 800
mm, decreasing to the north and west (Figure 2).
For the sediment study, we examined eight flood-control reservoirs and their watersheds
within the Lampasas River basin. Reservoirs L1, L2, L3, L4, L9, and LX are located in
Lampasas County and were constructed between 1958 and 1961. Before impoundment, the
parallel watersheds of L1, L2, and L3, contributed to the downstream watershed of LX.
Reservoirs M1 and M4, in Mills County, were completed in 1974. Basic attributes of the
reservoirs and their watersheds are compiled in Table 1.
Current local land use is predominantly rangeland, and livestock numbers have fluctuated
over the last several decades (Figure 3a) while remaining among the highest in the region
(Wilcox et al., 2012a). Cropland was widespread early in the 20[th] century (Figure 3b) but had
declined by nearly 80% by 2012 (Berg, M. D., manuscript in review, 2015). Amid this
shifting land use, the area has been characterized by large fluctuations in the extent of woody
plant cover, due to brush management and regrowth (Figure 3c), and a dramatic increase in
the density of farm ponds (Figure 3d) over the last several decades (Berg et al., 2015a).

## 2.2   Rainfall and runoff trends

To investigate local hydrological trends, we analyzed historical precipitation and streamflow
data for the Lampasas River basin. We created a composite record of annual precipitation
using a Thiessen polygon approach, centering polygons on available NWS stations (Figure 2).
Streamflow data were derived from the two USGS stream gage stations downstream from the
study watersheds. The lower Youngsport station, with a drainage area of 3,212 km[2], operated
between 1924 and 1980; the Kempner station, with a drainage area of 2,119 km[2] has remained
active from 1963 to the present.
We performed an automated baseflow separation of streamflow data from each station
(Arnold and Allen, 1999). This digital filter approach is objective and reproducible and
partitions annual baseflow and stormflow with high efficiency (Arnold et al., 1995)—
enabling these components to be interpreted in light of changing landscape conditions.
Using the precipitation and two streamflow datasets (1924—1980; 1963—2010), we applied a
nonparametric Mann-Kendall trend test to detect directional changes (Lettenmaier et al.,
1994). We performed two-tailed statistical tests for significance, with $\alpha = 0.10$.





## 2.3  Reservoir sedimentation rates
To shed light on sediment transport processes, we extracted cores from each of the eight
reservoirs and analyzed sediments using Cesium-137 ($^{137}$Cs) and Lead-210 ($^{210}$Pb) tracers.
$^{137}$Cs is present in the environment as a result of atomic weapons testing and accidental
emissions. $^{210}$Pb occurs naturally. Both can be used to estimate sedimentation rates and
interpret transport history in a variety of environments (Walling et al., 2003;Ritchie and
McHenry, 1990;Appleby and Oldfield, 1978). Coring sites were selected by locating the
thickest sediment deposits through exploratory hydroacoustic surveys (U.S. Army Corps of
Engineers, 2013, 1989;Dunbar et al., 2002). In each reservoir, we extracted sediment cores at
identified sites near the dam structure, from locations corresponding to the pre-impoundment
floodplain (Figure 4). Taking cores from these areas reduces the likelihood of capturing
mixed profiles, which skew analysis (Sanchez-Cabeza and Ruiz-Fernández, 2012). It also
ensures the collection of fine sediments, to which the radioisotopes preferentially adsorb
(Bennett et al., 2002). We extracted cores using a portable vibracoring system suspended from
a floating platform. This method captures unconsolidated, saturated sediments with minimal
disturbance and compaction (Lanesky et al., 1979). The cores were collected with an
aluminum pipe lowered to the point of refusal, penetrating the pre-impoundment surface.
Retrieved cores were sealed and transported upright to cold storage (~5°C).
We sectioned each core vertically in 3-cm intervals, drying each section for analysis
according to IAEA (2003) protocols. A subsample of each core section was ground to
homogenize its contents, sealed in a 50 mm x 9 mm Petri dish, and allowed to ingrow for at
least 21 days so that $^{210}$Pb supported levels reached equilibrium. Counts for $^{210}$Pb and $^{137}$Cs
were performed according to Hanna et al. (2014) using a Canberra low-energy germanium
gamma spectrometer. Radioisotope activity was indicated by photopeaks at 46 keV (total
$^{210}$Pb) and 661.6 keV ($^{137}$Cs). Excess $^{210}$Pb was calculated by subtracting the supported
activity of the $^{226}$Ra parent—obtained by averaging the 295, 351.9, and 609.3 keV peaks of
the $^{214}$Pb and $^{214}$Bi daughter products—from total measured $^{210}$Pb activity at the 46 keV peak.
Activity measurements were validated with IAEA-300 standard reference material.
To determine historical linear sedimentation rates, we used as a chronological marker the
depth of peak $^{137}$Cs activity (corresponding to the 1963 peak in global atmospheric fallout)
(Ritchie et al., 1973). We calculated average linear sedimentation rates for the post-1963
period by dividing this depth by the time elapsed between 1963 and the coring date for each





reservoir; we calculated the pre-1963 rates by dividing the depth of sediment below the
activity peak by the time elapsed between reservoir impoundment and 1963.
To complement [137]Cs analysis, we used excess [210]Pb activities to calculate the linear
sedimentation rate for each core (Krishnaswamy et al., 1971;Bierman et al., 1998). We also
searched for changing deposition rates within each core, as plots of the natural log of excess
[210]Pb versus depth indicate stable sedimentation rates over time when $R^2$ approaches 1.0.
Finally, we obtained historical annual Palmer Modified Drought Index (PMDI) data for the
region to identify potential climatic drivers of sedimentation during different periods. We
plotted PMDI and annual peak flows (from USGS data) between 1924 and 2010, identifying
episodes conducive to increased sediment production (in particular, a wet year or years
following a period of intense drought).
## 3    Results
### 3.1    Rainfall and runoff trends
Despite a great deal of interannual variability, there was no directional change in local
precipitation 1924—1980 ($p = 0.90$) or 1963—2010 ($p = 0.22$), which has remained near a
long-term average of 800 mm (Figure 5a). The same is true of total streamflow (1924—1980:
$p = 0.98$, 1963—2010: $p = 0.34$), which has averaged between 60 and 70 mm (Figure 5b). As
a result, the rainfall–runoff ratio also remained unchanged, at approximately 8% (1924—
1980: $p = 0.90$, 1963—2010: $p = 0.45$). Moreover, neither baseflow nor stormflow exhibited a
directional change over either period of record. However, baseflow as a proportion of total
streamflow did increase 1924—1980 ($p = 0.02$) despite minimal change in overall flow—
almost doubling its contribution (Figure 5c).
### 3.2    Reservoir sedimentation rates
Sediment core profiles varied widely in depth between reservoirs—from less than 3 cm in LX
to 162 cm in L1 (Figure 6). Activity peaks of [137]Cs supported the analysis of pre-1963
sedimentation rates for reservoirs L1, L2, L3, and L9. Overall, linear sedimentation rates were
higher before 1963 (Table 2; Figure 7). Except in the case of L3, sediment deposition has
slowed since 1963—by 54% in L1, 76% in L2, and 84% in L9. In reservoir L3, it increased



by 49% after 1963. Reservoir L1 exhibited the highest sedimentation rate both before and
after 1963. However, when normalized by catchment area, sedimentation rates varied much
more widely. That in L9 was by far the highest—surpassing the next highest reservoir by
nearly 1400% for the pre-1963 period and by 423% for the post-1963 period.
Cores from L4, LX, M1, and M4 did not display a $^{137}$Cs peak. For these cores, sedimentation
was assumed to be post-1963 and was estimated by dividing sediment depth by time since
impoundment. For cores L4 and M4, which did not capture the entire sediment profile, actual
rates likely are higher than those calculated.
Cores from reservoirs LX and M1 showed vertical mixing that prohibited $^{210}$Pb analysis.
However, remaining cores displayed high correlation between $^{210}$Pb activities and depth,
indicating linear sedimentation rates have remained quite stable over time (Table 2). $^{210}$Pb-
based estimates generally resembled those based on $^{137}$Cs activities. In addition, rates
calculated from $^{210}$Pb activities were similar to the post-1963 rates based on $^{137}$Cs activities ($p$
= 0.84), suggesting good agreement between the two methods for the period since 1963.
Chronological data revealed periods of drought of varying intensity and occasional years of
very high streamflow (Figure 8). The historic 1950s drought was longer and more severe than
any other over the last century; it was followed by periods of very high flow in 1957 and
1960. Comparable high flows in 1965 occurred in the middle of a multi-year drought, and the
severe drought beginning in 2006 featured occasional elevated peak flows. In 1992, very high
flows occurred during a prolonged wet period.

## 4   Discussion

### 4.1   Rainfall and runoff trends

Given the varying trends in precipitation and streamflow observed in many regions (Lins and
Slack, 1999;Andreadis and Lettenmaier, 2006), the dynamic hydrological stability in our
study area is surprising. At the same time, such consistency sheds light on the effects of
watershed changes on local water budgets. Studies at small spatial scales frequently indicate
that landscape changes have important water resource impacts, with the specific response
depending on the relative importance of evapotranspiration, recharge, and runoff (Foley et al.,
2005;Kim and Jackson, 2012). Such changes affect local water budgets and influence water





yields (Petersen and Stringham, 2008;Huxman et al., 2005;Farley et al., 2005). However,
complicated feedbacks make effects at larger scales highly uncertain and often overwhelmed
by climatic and physical characteristics (Peel, 2009;Wilcox et al., 2006;Kuhn et al., 2007).
Our rainfall–runoff ratio of 8% is essentially identical to early estimates of 7% for the area
(Tanner, 1937). The lack of a directional trend in streamflows suggests that this region, like
many semiarid landscapes dominated by surface runoff, is largely hydrologically insensitive
to shifting watershed characteristics (Wilcox, 2002). Changes in land use and land cover—
and even the impoundment of small reservoirs—have had negligible impacts on streamflow.
It is still not understood why baseflow showed a proportional increase 1924—1980. In some
landscapes, improving range conditions have led to increased infiltration (Wilcox and Huang,
2010). However, local livestock numbers have remained high, and karst features are limited—
unlike other regions where baseflow increases have been attributed to rangeland recovery. It
is possible that infiltration from local impoundments has added to baseflows. Despite minimal
effects on total streamflow, even small dams can create localized groundwater recharge (Graf,
1999;Smith et al., 2002), and Lampasas River tributaries are characterized by a high degree of
connectivity between surface water and local aquifers (Mills and Rawson, 1965).
Perennial flow in this part of the Lampasas River is maintained by isolated springs fed by an
aquifer extending beyond the basin (Mills and Rawson, 1965). As a result, the effective
catchment of the river is larger than it appears, and springflow contributions complicate the
interpretation of streamflows. At the same time, it is clear that the fundamental relationship
between rainfall and streamflow has not changed over more than 85 years—suggesting that
the Lampasas River is hydrologically resilient in the face of changing land use and land cover.
**4.2   Reservoir sedimentation rates**
Because sediment deposition affects reservoir storage and flood detention, understanding
sedimentation rates over time is critical to managing rangeland water resources. Though
questions do remain regarding the opposing trend in reservoir L3, changes in rates make it
clear that sedimentation was more rapid before 1963. The period since that time has been
characterized by stable and lower yields. But what explains the higher rates seen during the
earlier period? Additional historical landscape data may offer a key interpretive lens.
Livestock can be powerful instruments of landscape change, both directly (trampling soils)
and indirectly (disturbing protective vegetation). When grazing is prolonged or intense,



sediment yield can be great (Trimble and Mendel, 1995). The high animal densities in this
area around the time of reservoir impoundment doubtless contributed to erosion (Figure 3a).
Crop production also can result in accelerated erosion by damaging soil structure and
depleting organic matter (Quine et al., 1999). Cropland is a major source of sediment in many
landscapes (Foster and Lees, 1999;Blake et al., 2012). In our study area, cropland acreage has
declined dramatically since the 1930s (Figure 3b). Further, nationwide improvements in soil
conservation have reduced sediment yield from many agricultural lands (Knox, 2001).
While woody plant encroachment influences soil loss, removing undesirable shrubs and trees
also elevates short-term sediment yields (Porto et al., 2009). Since the time of initial
settlement, woody plant management has resulted in major land cover changes (Figure 3c).
Most early removal was done manually, and the first mechanical control methods were very
destructive, leading to high erosion rates (Hamilton and Hanselka, 2004). In recent decades,
however, brush removal has declined with shifting landowner priorities (Sorice et al., 2014).
Changes in precipitation frequency, duration, or intensity also affect sediment transport (Xie
et al., 2002;Allen et al., 2011). Similarly, drought is an important driver of sediment dynamics
in many rangelands. Extended dry periods can cause long-term shifts in plant cover, leading
to sediment pulses when rains return (Allen and Breshears, 1998;Nearing et al., 2007). The
Lampasas River experienced very high flows in 1957, 1960, 1965, and 1992, and some of
these were associated in time with severe droughts (Figure 8). Just before the impoundment of
most of the reservoirs we examined, the region was in the grip of drought conditions
unmatched since European settlement (Bradley and Malstaff, 2004). Our sediment records
cover only the end of this drought but show pre-1963 deposition 220–630% faster than
subsequent rates. However, any direct effects of the 1957 drought-breaking floods would not
be found in the sediments of the reservoirs, which were impounded beginning in 1958.
Interestingly, we also did not find spikes in sedimentation associated with high flows or
droughts later in the study period. The apparent low importance of drought and floods in
sediment delivery in these watersheds is surprising.
Together, these factors have acted over multiple temporal and spatial scales to influence
sediment yields in the study area. Yet because there is no clear link between contemporary
land use, land cover, and sedimentation rates, it is possible that another process has reduced
sediment yields.




## 4.3 Sediment storage

To truly understand the local sediment processes at work, it is important to understand what our findings actually show. Sedimentation rates are poor indicators of in-field soil erosion and redistribution (Nearing et al., 2000;Ritchie et al., 2009); what they do reflect is more closely related to net watershed sediment yield. Sediment yield is buffered by internal storage. Especially at larger scales, watersheds can have a great deal of internal storage, so that very little eroded soil actually leaves the watershed, even in the presence of extreme erosion (Bennett et al., 2005;Porto et al., 2011).

In this study area, the increasing density of farm ponds (Figure 3d) represents a key potential sink for watershed sediments. These ponds retain material that otherwise would be transported downstream, reducing sediment yields. Because of their smaller contributing watersheds, ponds have high trap efficiencies, magnifying their effects (Brainard and Fairchild, 2012). Indeed, impoundments may be the single greatest anthropogenic modifier of sediment transport; globally, most sedimentation now takes place in aquatic settings and will be retained therein for long periods (Renwick et al., 2005;Verstraeten et al., 2006).

In addition to this storage of eroded sediments in local ponds, a vast amount of sediment from past erosion likely remains on the landscape (Beach, 1994;Meade, 1982). The initial decades after European settlement in this area saw intensive cultivation and very high livestock densities (Jordan-Bychkov et al., 1984;Wilcox et al., 2012a). This destructive combination remained in place for nearly a century in the Lampasas Cut Plain. By the 1930s, many rangelands were already seriously degraded (Mitchell, 2000;Bentley, 1898;Box, 1967). While the methods we used do not allow us to determine whether reservoir sediments result from contemporary erosion or are a legacy of earlier land use, stabilizing sediment yields and observations of local gully erosion suggest that deposits from prior erosion continue to be a source of sediment (Bartley et al., 2007;Mukundan et al., 2011;Phillips, 2003).

The lack of sediments in LX appears to lend support to the importance of internal deposits. This reservoir's watershed is comparable in size to those of L2, L3, and M4, yet sedimentation rates were only 3%–14% of those in the other reservoirs. When L1, L2, and L3 were impounded, the effective catchment area of LX decreased by 86%. Without the historical streamflows and sediment loads from those tributaries, deposits are no longer mobilized and transported downstream.





Given this complexity, we suggest that radioisotope tracers have great potential to elucidate
the dynamics of rangeland systems, particularly as their use evolves from primarily research
applications to use as a management and decision-support tool (Mukundan et al., 2012).
Further strides can be made in understanding rangeland processes by (1) incorporating
historical climate, land use, and land cover information to interpret sediment data (Venteris et
al., 2004;Boardman, 2006) and (2) including sediment surveys of the farm ponds that are
much smaller yet far more abundant than the reservoirs we examined (Downing et al., 2006).
**5    Conclusion**
We examined long-term trends in rainfall, runoff, and sediment yield in rangeland watersheds
with a dynamic land use history. Over more than 85 years, neither precipitation nor
streamflow showed any directional trend, suggesting a lack of hydrological sensitivity to
landscape change. This raises doubts over efforts to increase runoff by directing land cover
changes. Reservoir sedimentation rates generally were higher before 1963, and then stabilized
at a lower level over the 50 years since 1963. We believe that this decline in sediment yield is
related to long-term landscape changes and an increase in internal storage. As a result, future
changes in land use or sediment storage may impact downstream reservoir capacity. These
findings challenge simplistic assumptions about streamflow and sediment yield in dynamic
rangelands. Determining the role of these landscapes in meeting growing water resource
demands requires a creative approach. Integrating multiple techniques with historical
information enables a more complete understanding of rangeland processes and holds the key
to informed water planning.
**Data availability**
Streamflow data are available at the USGS National Water Information System. Stream
gages: 08103800 (Kempner) and 08104000 (Youngsport). Drought data are available at the
NOAA National Climate Data Center. Texas Climate Division: CD 3 (North Central) and CD
6 (Edwards Plateau).
**Acknowledgements**



Dan Duncan, Andrea Hanna, and Diana di Leonardo performed activity counts of sediment
samples. This work was supported by USDA-NIFA Managed Ecosystems grant 2011-68002-
30015, USDA-NIFA National Needs Program grant 2009-38420-05631, NSF-CNH grant
413900, and a Tom Slick Graduate Research Fellowship from the Texas A&M University
College of Agriculture and Life Sciences.



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





1    Table 1. Sediment study reservoirs and watershed characteristics.

| Reservoir | Primary Inflow | Surface Area (km$^2$) | Watershed Area (km$^2$) | Year Impounded | Year Cored | Min. Elev. (m) | Max. Elev. (m) |
|---|---|---|---|---|---|---|---|
| L1 | Donalson Creek | 0.20 | 50.9 | 1959 | 2010 | 367 | 500 |
| L2 | Pitt Creek | 0.18 | 23.2 | 1959 | 2010 | 362 | 458 |
| L3 | Espy Branch | 0.11 | 27.5 | 1958 | 2010 | 355 | 459 |
| L4 | Pillar Bluff Creek | 0.07 | 41.2 | 1960 | 2012 | 345 | 467 |
| L9 | Cemetery Creek | 0.02 | 1.2 | 1960 | 2012 | 322 | 363 |
| LX | Bean Creek | 0.20 | 23.1 | 1961 | 2012 | 338 | 420 |
| M1 | Middle Bennett Creek | 0.14 | 34.6 | 1974 | 2012 | 422 | 536 |
| M4 | Mustang Creek | 0.15 | 28.0 | 1974 | 2012 | 432 | 534 |





Table 2. Linear sedimentation rates derived from radioisotope activities.

| Core | $^{137}$Cs | | | | $^{210}$Pb | | |
| | Pre-1963 | | Post-1963 | | Core mean | | $R^2$ |
| | cm y$^{-1}$ | cm y$^{-1}$ km$^{-2}$ | cm y$^{-1}$ | cm y$^{-1}$ km$^{-2}$ | cm y$^{-1}$ | cm y$^{-1}$ km$^{-2}$ | ln dpm g$^{-1}$ |
|---|---|---|---|---|---|---|---|
| L1 | 6.4 | 0.13 | 2.9 | 0.06 | 3.1 | 0.06 | 0.90 |
| L2 | 3.4 | 0.15 | 0.8 | 0.03 | 0.9 | 0.04 | 0.97 |
| L3 | 1.4 | 0.05 | 2.1 | 0.08 | 1.3 | 0.04 | 0.96 |
| L4 | a | a | 0.5[b] | 0.01[b] | 1.2 | 0.03 | 0.93 |
| L9 | 2.5 | 2.02 | 0.4 | 0.32 | 0.4 | 0.19 | 0.94 |
| LX | a | a | 0.1 | < 0.01 | c | c | c |
| M1 | a | a | 1.5 | 0.04 | c | c | c |
| M4 | a | a | 0.4[b] | 0.01[b] | 0.8 | 0.01 | 1.00 |

[a]Core did not display a $^{137}$Cs peak, and rates were calculated using the time elapsed since
impoundment.
[b]Core did not capture the pre-impoundment surface and likely underestimates true values.
[c]Core showed significant vertical mixing, preventing calculation of sedimentation rate.





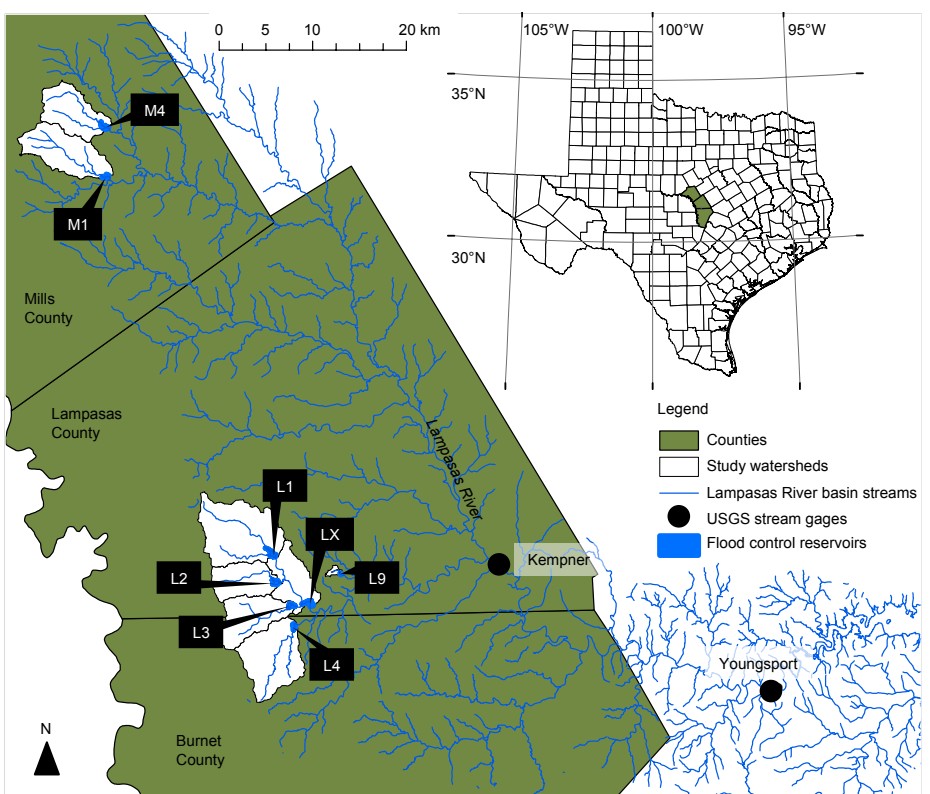

2    Figure 1. Study area in Texas, USA. Each study watershed encloses a flood control reservoir

3    from which sediment cores were collected. All watersheds contribute flow to the Lampasas

4    River.



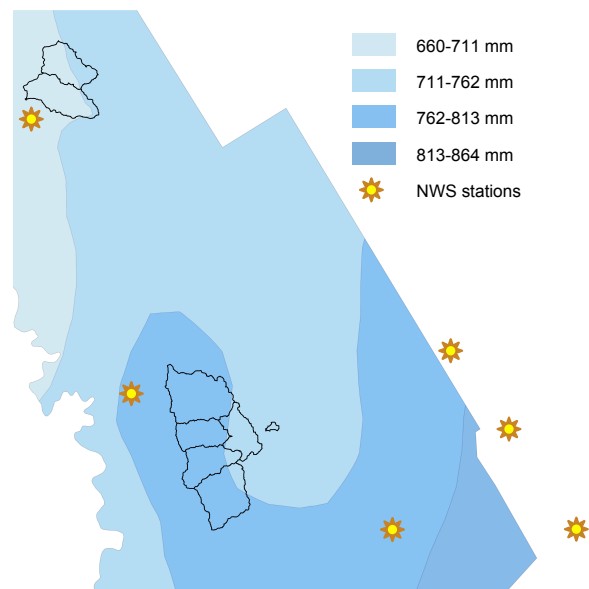

2    Figure 2. Average annual precipitation gradient and location of National Weather Service

3    (NWS)    stations    used    to    construct    historical    precipitation    record.



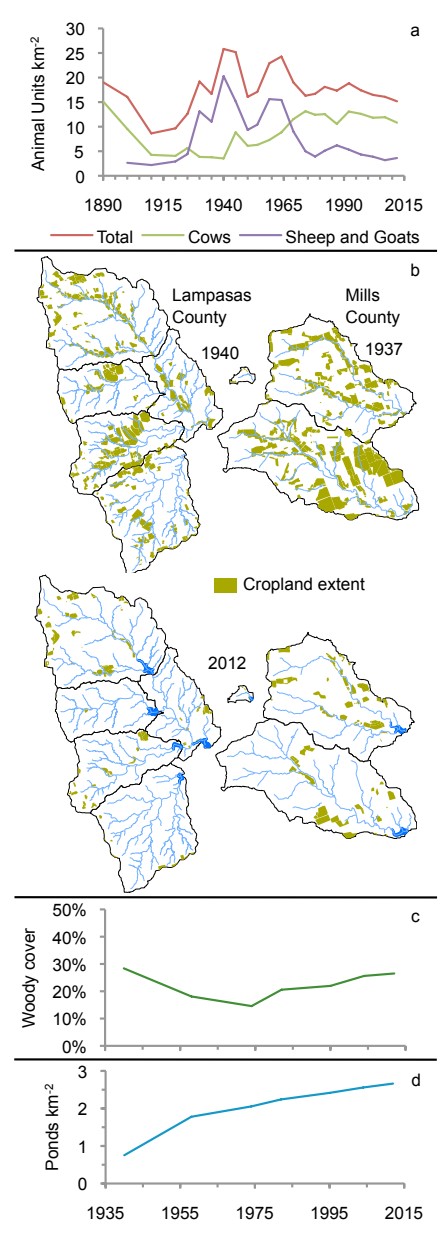

Figure 3. Historical landscape changes in the study area. (a) Livestock numbers in the Lampasas Cut Plain. Recreated from Wilcox et al. (2012a). (b) Extent of active cropland in 1937-40 and 2012 (Berg, M. D., manuscript in review, 2015). (c) Historical extent of woody plant cover in the study watersheds **(**Berg et al., 2015b**)**. (d) Pond density over time in the study watersheds (Berg et al., 2015a).



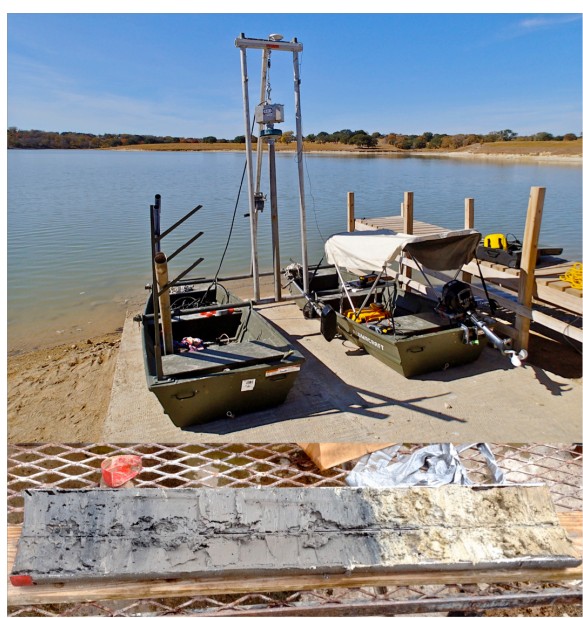

2    Figure 4. Reservoir sediment coring apparatus (top) and representative sediment profile

3    (bottom).



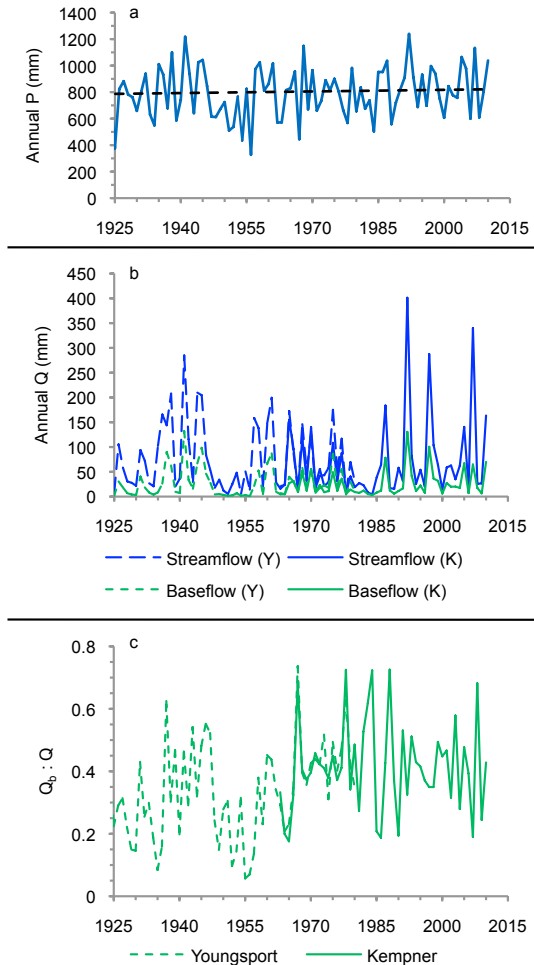

Figure 5. Precipitation and streamflow trends of the Lampasas River basin. (a) Precipitation
showed no directional trend. (b) Streamflow showed no directional trend at either the
Youngsport (Y) or Kempner (K) station, despite being highly variable. (c) Baseflow as a
proportion of total streamflow displayed an upward trend over the first portion of the study
period.





Figure 6. Sediment core profiles of bulk density and radioisotope activities from the eight
reservoirs. Solid horizontal lines indicate the pre-impoundment surface (no line indicates the
core did not capture the pre-impoundment surface). Dashed lines in [137]Cs graphs represent the
depth of peak activity. The [210]Pb profile for L3 is from a second core collected at the same
location.





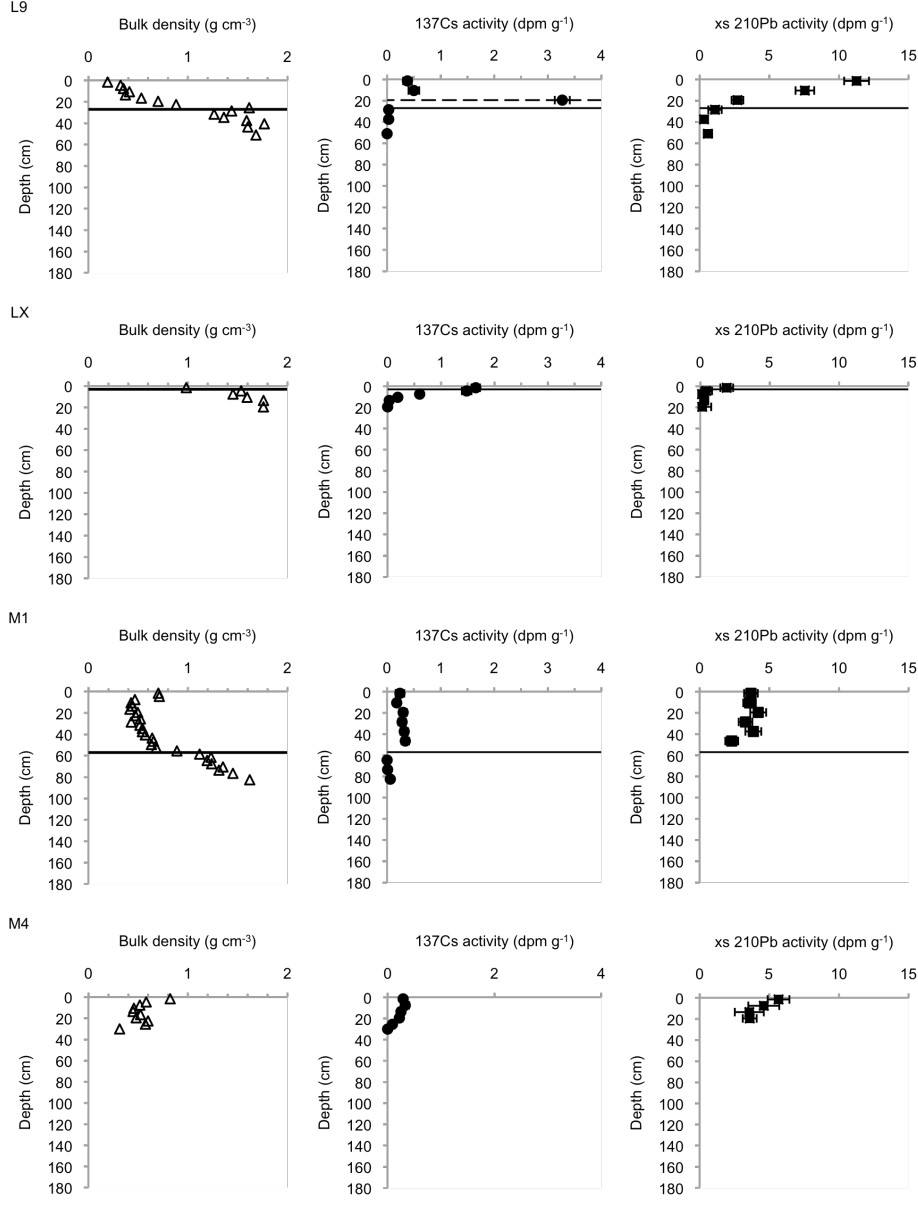

Figure 6 (continued). Sediment core profiles of bulk density and radioisotope activities from
the eight reservoirs. Solid horizontal lines indicate the pre-impoundment surface (no line
indicates the core did not capture the pre-impoundment surface). Dashed lines in $^{137}$Cs graphs
represent the depth of peak activity.





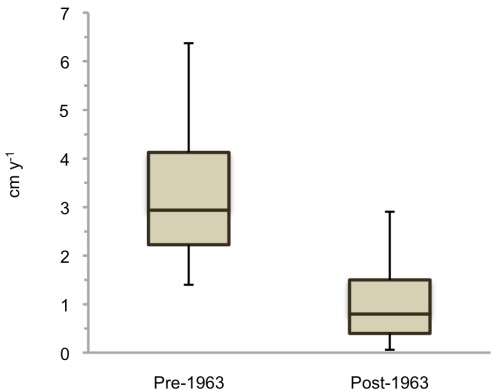

2    Figure 7. Linear sedimentation rates derived from [137]Cs activities. Summary comparison of

3    pre-1963 and post-1963 rates.





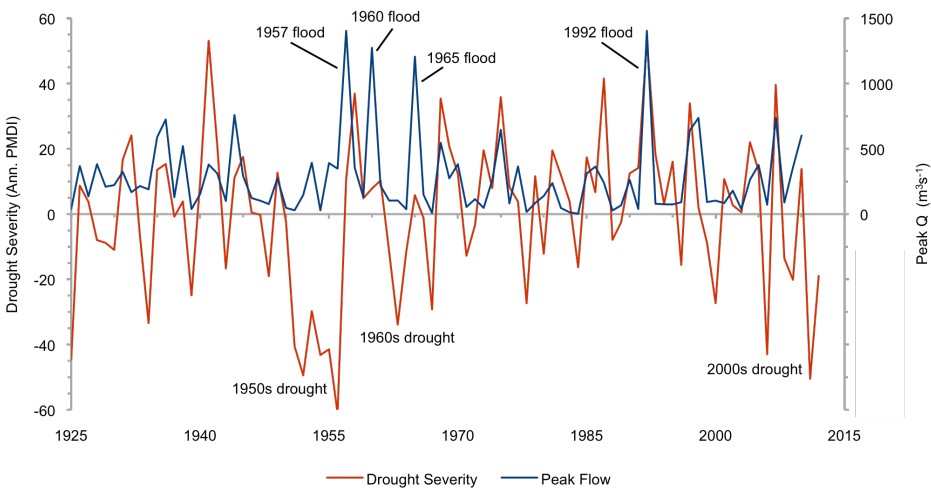

2    Figure 8. Chronology of regional drought (annual Palmer Modified Drought Index) and peak

3    flows on the Lampasas River.