# Peer review of "Contrasting watershed-scale trends in runoff and sediment"

_Hydrology and Earth System Sciences, 2015_

## Referee Comment (RC1) · Anonymous Referee #1 · 26 Jan 2016

Review of Berg et al. – submitted to Hydrology and Earth System Sciences Berg et al. presents reservoir sedimentation data from a series of watersheds in Central Texas and integrates it with long-term precipitation and streamflow data to evaluate the impact of landscape scale changes on water resources. Long-term hydrologic studies that span multiple scales are of high interest to readers of HESS. However, the study has some significant shortcomings that counteract its main message of relatively stable precipitation-runoff relationships despite landscape scale changes. In particular, some of the data presented are less effective in addressing the central questions while other important data are missing. These are outlined in the main comments below. 1. Evaluation of type of vegetation change or changes in potential ET on rainfall-runoff

relationships. Given that long-term runoff (Q) is considered as the difference between precipitation and evapotranspiration (Q=P-ET), the authors spend little time evaluating the potential impacts of temperature changes or plant rooting depth on changing ET and Q. If the actual vegetation changes did not result in a change in rooting depth or the length of the plant growing season, then we would logically not expect any change in Q. However if rooting depth and/or growing season length decreased while potential ET increased, then these changes would offset and ET would stay the same. The relationship between these parameters is the most important for predicting rainfall-runoff relationships in the future, yet no data are presented on these variables. Potential ET can be relatively easily calculated from the PRISM data (Oregon State), and the rooting depth and season length could be assigned to each cover type (woody cover, grassland, and crop land). 2. There is no discussion of the impact of slope on sedimentation. I know you focus more on relative changes across time, but I think including the discussion of slope impacts is particularly important, especially with respect to internal sediment storage from on farm ponds. 3. With respect to the baseflow analysis, it would be extremely helpful to know the average pond size across time (Fig. 3). Is it 0.1 hectares, 1 hectare, larger? Moreso than just the pond density, knowing the average size is critical for assessing potential baseflow contributions from pond recharge and reduction in overland runoff. Specific comments: Section 2.1. Would be good to report mean air T, RH, and factors that affect PET. Page 4, line 11: Would be good to included characteristic rooting depths and growing periods for each of these types of vegetation. Page 4, line 16: Would be good to include potential ET in this section. Table 1: Would be extremely useful to have a pond density/average pond size and average upstream slope data for this table. Figure 4: Not sure this figure is needed. Figure 7: This graph seems to duplicate Table 2. I would recommend adding rows to Table 2 with the data from this figure presented there.

---

## Referee Comment (RC2) · Anonymous Referee #2 · 5 Apr 2016

Historical pattern of hydrological behaviors in the context of climate change and human disturbance is a hot topic not only for retrospectively knowing the past and capture the present, but also for planning for the future. It is an interesting paper to test long-term trend of water and sediment yield at watershed scale in rangeland landscapes, and explore the effect of landscape changes. A variety of data sources were generated and interpreted at various temporal and spatial scales, but it also represent main aspects that need to be improved before it can be considered for publication. Therefore, I suggest a substantial major revision.

Firstly, above the two hydrological stations where streamflow data were applied in this paper, whether there are dams? Of course there must be. Consequently, annual

streamflow data for analysis was inadequate because dams may repartition streamflow on seasonal basis. So, monthly streamflow data is needed to be further analysed. I deduce the hydrological insensitivity cannot be directly deduced from the non-directional trend of annual streamflow, maybe the precipitation-water relationship (not just runoff-rainfall ratio) is useful for analysis. Also, dams may mask the hydrological effect of landscape change on water yield at relatively large spatial scales.

Secondly, sediment yield is scale-dependent. It can be both changed at first-order watersheds by landscape change and at larger catchments by integration of landscape change and alteration of fluvial hydrological connectivity (e.g., dam construction). Cs-137 dating provides one time-marker (Cf. 1963) to separate the profile and reflecting sediment dynamics in first-order watersheds, but significant landscape change is consistent with this marker? Also, dating results can not reflect sediment status at downstream hydrological stations, which were not comparable relative to water yields at downstream hydrological stations.

Thirdly, of course, all these factors mentioned may be responsible for temporal changes of sediment dynamics. Interpretation and analysis can be carried out in more deep and specific way.

Detailed points: Page 4, line 17-19: where did the precipitation data come from? and its temporal length? Are they county-level average or watershed average? Page 6, line 19: runoff-rainfall ratio? Table 1: please add a column to indicate the core length or compaction factor for each core extracted from the individual reservoirs.

---

## Author Response (AR1)

We thank the editor and the reviewers for helping refine our work. The time and effort is always greatly appreciated, and it helps us clarify our message to make the greatest impact for the hydrological community. Some points and questions from the reviewers do give us pause, yet we have made changes to address these. Some comments by Reviewer #1 indicate a desire to evolve the present study into a new project that is well beyond our stated objectives. This is perhaps underpinned by an expectation of climate change impacts, which have actually not been borne out locally. Comments by Reviewer #2 suggest a lack of clarify, for which we apologize and have made the appropriate tweaks to improve readability. Finally, to address comments from the editor regarding depth/source of water by woody plants, we have included new text with relevant citations in section 4.1. With these changes outlined below, we are even more confident that this work is ready for a wider audience and for these findings to be quickly injected into water planning efforts.

Sincerely,
Matthew Berg and co-authors

**Response to Reviewers**

Reviewer #1
*Berg et al. presents reservoir sedimentation data from a series of watersheds in Central Texas and integrates it with long-term precipitation and streamflow data to evaluate the impact of landscape scale changes on water resources. Long-term hydrologic studies that span multiple scales are of high interest to readers of HESS.*

We appreciate this perspective and agree with the reviewer that HESS makes an ideal fit for a finalized manuscript describing our work. We internally have gone through many iterations of this paper and believe it makes an important contribution to a key ecohydrological question. We thank Reviewer #1 for time and effort in providing input toward this end.

*1. Evaluation of type of vegetation change or changes in potential ET on rainfall-runoff relationships. Given that long-term runoff (Q) is considered as the difference between precipitation and evapotranspiration (Q=P-ET), the authors spend little time evaluating the potential impacts of temperature changes or plant rooting depth on changing ET and Q. If the actual vegetation changes did not result in a change in rooting depth or the length of the plant growing season, then we would logically not expect any change in Q. However if rooting depth and/or growing season length decreased while potential ET increased, then these changes would offset and ET would stay the same. The relationship between these parameters is the most important for predicting rainfall-runoff relationships in the future, yet no data are presented on these variables. Potential ET can be relatively easily calculated from the PRISM data (Oregon State), and the rooting depth and season length could be assigned to each cover type (woody cover, grassland, and crop land).*

We acknowledge the importance of precipitation and evapotranspiration to long-term streamflow and other components of runoff. However, one objective of this study was to quantify historical changes in streamflow. The important findings of our study are that neither precipitation nor streamflow has shown a directional trend over the study period.

This core question, with respect to streamflow, addresses a high-level conversation in the ecological community on the outcomes of woody plant encroachment of rangelands. As such, we examined the aggregate of multiple potential impacts as actual, direct measurements of streamflow. If neither precipitation nor streamflow has changed, the net changes in landscape-scale components of ET are also insignificant.

Further emphasizing this point are the following: (1) since long-term temperature changes have not been seen in this area (Meehl et al., 2012), temperature is not of primary concern here and (2) much of this region is characterized by shallow soils and karst geology that limit root penetration for nearly all species.

Clearly, field studies of historical changes in rooting depth are not possible given the timeframe we examined. However, an abundance of work by Susan Schwinning and others has indicated that the dominant woody plant species in this region do not access deep storage. The evergreen nature of these woody plant species and their allegedly enormous transpiration capacities have been put forward as reasons why woody plant removal should theoretically yield streamflow increases. This faulty rationale persists in decision-making processes and demands clarification of hydrological processes.

Evidence suggests it is precisely because ET is high, that woody plants are disconnected from deeper storage, and that assumptions of transpiration characteristics are unfounded, that local increases in woody vegetation will not appreciably affect runoff. As such, we focused on streamflow, interpreted with precipitation. This is the critical information that is needed in a timely manner. New text to this effect has been added in 4.1.

*2. There is no discussion of the impact of slope on sedimentation. I know you focus more on relative changes across time, but I think including the discussion of slope impacts is particularly important, especially with respect to internal sediment storage from on farm ponds.*

The region is characterized by 1-5% slopes for nearly all soils, and channel beds are rock. As the reviewer implies, while dams may have caused minor changes to stream channel slopes in very localized areas, neither average channel slopes nor upland slopes have changed.

*3. With respect to the baseflow analysis, it would be extremely helpful to know the average pond size across time (Fig. 3). Moreso than just the pond density, knowing the average size is critical for assessing potential baseflow contributions from pond recharge and reduction in overland runoff.*

Nearly all ponds are less than 0.3 ha when full immediately after rain events. Pond size over time was not included because dramatic seasonal and interannual fluctuations in depth and area due to climatic variability make these data of extremely limited value. As water levels are unregulated, a single pond can cease to hold water in summer and then overflow in wet periods within the same year. Hydrological – and sediment – impacts from these features are more directly due to the impoundment of intermittent streams behind dams themselves. As a result, while the size of ponds may occasionally play a role in localized hydrological impacts, the extreme variability of local storage by intermittent streams in a semiarid region makes the number of dams/ponds of much greater importance.

*Specific comments:*
*Section 2.1. Would be good to report mean air T, RH, and factors that affect PET.*

For the reasons above, we elected not to include these secondary variables. This region is located in what's been called the ''Warming Hole'' in the southern United States, where long-term temperature increases have not been experienced. Nonetheless, to enhance the site description and provide the reviewer a little better feel for PET factors, we have included average temperatures for both winter and summer. RH is tremendously variable, with this region located at the convergence of dry continental and moist Gulf of Mexico air masses.

*Page 4, line 11: Would be good to included characteristic rooting depths and growing periods for each of these types of vegetation.*

Again, since these were beyond the scope of this study and that the plant communities here are complex and limited by shallow soils, we elected not to include these data.

*Figure 4: Not sure this figure is needed.*

We believe this figure adds to the reviewers' ability to visualize our methodology, and internal reviews have consistently made suggestions to retain it.

*Figure 7: This graph seems to duplicate Table 2. I would recommend adding rows to Table 2 with the data from this figure presented there.*

Though not duplicative, we do recognize that Figure 7 is derived from the same data as Table 1. However, we feel that a separate Figure 7 illustrates in striking fashion the different linear sedimentation rates over time. Since this is one of two main objectives of this study, we feel it has an important place in the manuscript.

Reviewer #2
*Historical pattern of hydrological behaviors in the context of climate change and human disturbance is a hot topic not only for retrospectively knowing the past and capture the present, but also for planning for the future. It is an interesting paper to test long-term trend of water and sediment yield at watershed scale in rangeland landscapes, and*

*explore the effect of landscape changes. A variety of data sources were generated and interpreted at various temporal and spatial scales, but it also represent main aspects that need to be improved before it can be considered for publication. Therefore, I suggest a substantial major revision.*

We appreciate the time and effort taken by Reviewer #2 and heartily agree about the importance of hydrology and human impact. However, the key focus of this paper is not on climate change (see comment above in response to Reviewer #1) but rather on the potential impacts of dramatic changes in land use and land cover. We have made some changes to highlight that importance, but we cannot identify any specific recommended changes that would warrant "substantial major revision." We are confident that our improvements to clarify explanation in the manuscript have accounted for essentially all of the questions posed by reviewer #2

*Firstly, above the two hydrological stations where streamflow data were applied in this paper, whether there are dams? Of course there must be. Consequently, annual streamflow data for analysis was inadequate because dams may repartition streamflow on seasonal basis. So, monthly streamflow data is needed to be further analysed. I deduce the hydrological insensitivity cannot be directly deduced from the non-directional trend of annual streamflow, maybe the precipitation-water relationship (not just runoff-rainfall ratio) is useful for analysis. Also, dams may mask the hydrological effect of landscape change on water yield at relatively large spatial scales.*

There are in fact no other dams on the Lampasas River above the streamflow gage stations. Smaller dams on intermittent tributaries mentioned in this paper are for flood control purposes, retaining small volumes only temporarily. In addition, we did analyze streamflow collected on a daily basis, and text has been adjusted to reflect this fact.

*Secondly, sediment yield is scale-dependent. It can be both changed at first-order watersheds by landscape change and at larger catchments by integration of landscape change and alteration of fluvial hydrological connectivity (e.g., dam construction). Cs-137 dating provides one time-marker (Cf. 1963) to separate the profile and reflecting sediment dynamics in first-order watersheds, but significant landscape change is consistent with this marker? Also, dating results cannot reflect sediment status at downstream hydrological stations, which were not comparable relative to water yields at downstream hydrological stations.*

We absolutely agree and appreciate this perspective. The reviewer will note that Table 2 includes columns that describe per-area sediment deposition data for both Cs-127 and Pb-210 to account for this very issue. We took this step to normalize data and allow for stronger comparisons among watersheds. In addition, the inclusion of $^{210}$Pb supports a moving window of analysis that assesses changes over time rather than relying on comparison with a single chronological event. While the sediment data used here were collected from different locations than the streamflow gages, we believe this is actually a significant strength that increases our spatial resolution eightfold, providing previously unavailable high-resolution data for eight different watersheds rather than one location that represents the upper river basin.

*Thirdly, of course, all these factors mentioned may be responsible for temporal changes of sediment dynamics. Interpretation and analysis can be carried out in more deep and specific way.*

The authors very much agree. That is why these multiple factors have been included. Yet since they are intrinsically linked in time, with greatest impacts many decades in the past, the ability to identify with absolute certainty the driving factors in play at a specific point in time simply does not exist. In fact, the approach we have provided here – linking historical streamflow and sediment data in the context of landscape change over time – is novel, especially within important rangeland watersheds. The conclusions from this paper are relevant right now in regional water planning efforts and for answering pressing questions of how projections of water availability are made for these areas.

*Detailed points: Page 4, line 17-19: where did the precipitation data come from? And its temporal length? Are they county-level average or watershed average?*

These questions are answered in section 2.2. We did clarify with new language that the National Weather Service data covered the same date range as the USGS streamflow data.

*Page 6, line*
*19: runoff-rainfall ratio?*

Rainfall-runoff ratio is a common metric used for determining the proportion of precipitation in a watershed which leaves as streamflow. This is of utmost importance for this particular region, as semiarid rangeland watersheds increasingly are often relied upon as source water areas, these areas are poorly quantified, and most planning efforts focus on annual supplies and volumes. Agreement from our colleagues in the field suggests that this is an appropriate focal point of this paper. We do recognize that this was not explicitly defined and introduced new text in the manuscript toward that end.

*Table 1: please add a column to indicate the core length or compaction factor for each core extracted from the individual reservoirs.*

An indication of core length is included in Figure 6. We agree with Reviewer #1 that redundancy between tables and figures is not necessary and detracts from the presentation, so we have opted for a more visual approach of length alongside other related data.

[revised manuscript text omitted]
|------|------|------|------|------|------|------|------|
| L1 | 6.4 | 0.13 | 2.9 | 0.06 | 3.1 | 0.06 | 0.90 |
| L2 | 3.4 | 0.15 | 0.8 | 0.03 | 0.9 | 0.04 | 0.97 |
| L3 | 1.4 | 0.05 | 2.1 | 0.08 | 1.3 | 0.04 | 0.96 |
| L4 | a | a | 0.5[b] | 0.01[b] | 1.2 | 0.03 | 0.93 |
| L9 | 2.5 | 2.02 | 0.4 | 0.32 | 0.4 | 0.19 | 0.94 |
| LX | a | a | 0.1 | < 0.01 | c | c | c |
| M1 | a | a | 1.5 | 0.04 | c | c | c |
| M4 | a | a | 0.4[b] | 0.01[b] | 0.8 | 0.01 | 1.00 |

[a]Core did not display a $^{137}Cs$ peak, and rates were calculated using the time elapsed since impoundment.

[b]Core did not capture the pre-impoundment surface and likely underestimates true values.

[c]Core showed significant vertical mixing, preventing calculation of sedimentation rate.

[Figure]

Figure 1. Study area in Texas, USA. Each study watershed encloses a flood control reservoir from which sediment cores were collected. All watersheds contribute flow to the Lampasas

River.

[Figure]

660-711 mm
711-762 mm
762-813 mm
813-864 mm
☀ NWS stations

Figure 2. Average annual precipitation gradient and location of National Weather Service (NWS) stations used to construct historical precipitation record.

[Figure]

Figure 3. Historical landscape changes in the study area. (a) Livestock numbers in the

Lampasas Cut Plain. Recreated from Wilcox et al. (2012a). (b) Extent of active cropland in

1937-40 and 2012 (Berg, M. D., manuscript in review, 2015). (c) Historical extent of woody plant cover in the study watersheds **(**Berg et al., 2015b**)**. (d) Pond density over time in the study watersheds (Berg et al., 2015a).

[Figure]

Figure 4. Reservoir sediment coring apparatus (top) and representative sediment profile (bottom).

[Figure]

Figure 5. Precipitation and streamflow trends of the Lampasas River basin. (a) Precipitation showed no directional trend. (b) Streamflow showed no directional trend at either the Youngsport (Y) or Kempner (K) station, despite being highly variable. (c) Baseflow as a proportion of total streamflow displayed an upward trend over the first portion of the study period.

[Figure]

Figure 6. Sediment core profiles of bulk density and radioisotope activities from the eight reservoirs. Solid horizontal lines indicate the pre-impoundment surface (no line indicates the core did not capture the pre-impoundment surface). Dashed lines in [137]Cs graphs represent the depth of peak activity. The [210]Pb profile for L3 is from a second core collected at the same location.

[Figure]

Figure 6 (continued). Sediment core profiles of bulk density and radioisotope activities from the eight reservoirs. Solid horizontal lines indicate the pre-impoundment surface (no line indicates the core did not capture the pre-impoundment surface). Dashed lines in [137]Cs graphs represent the depth of peak activity.

[Figure]

Figure 7. Linear sedimentation rates derived from [137]Cs activities. Summary comparison of pre-1963 and post-1963 rates.

[Figure]

Figure 8. Chronology of regional drought (annual Palmer Modified Drought Index) and peak flows on the Lampasas River.